# Agroclimatic and Phytosanitary Events and Emerging Technologies for Their Identification in Avocado Crops: A Systematic Literature Review

Tomas Ramirez-Guerrero [1,*], Maria Isabel Hernandez-Perez [1], Marta S. Tabares [1], Alejandro Marulanda-Tobon [2] and Eduart Villanueva [3] and Alejandro Peña [3]

1 GIDITIC, Universidad EAFIT, Medellín 050022, Colombia; mihernandp@eafit.edu.co (M.I.H.-P.); mtabares@eafit.edu.co (M.S.T.)
2 GEMA, Universidad EAFIT, Medellín 050022, Colombia; amarula2@eafit.edu.co (A.M.-T.)
3 Información y Gestión, Universidad EAFIT, Medellín 050022, Colombia; evillanu@eafit.edu.co (E.V.); japena@eafit.edu.co (A.P.)
* Correspondence: teramirezg@eafit.edu.co

**Abstract:** Avocado is one of the most commercialized and profitable fruits in the international market. Its cultivation and production are centered in countries characterized by tropical and subtropical climatic conditions, many of them with emerging economies. Moreover, the use of technology is key to agricultural production improvement strategies. Using avocado crop data to forecast the potential impacts of biotic and abiotic factors, combined with smart farming technologies, growers can apply measures during a single production phase to reduce the risks caused by pests and weather variations. Therefore, this paper aims to distinguish the most relevant variables related to agroclimatic and phytosanitary events in avocado crops, their incidence on production and risk management, as well as the emerging technologies used for the identification and analysis of pests and diseases in avocados. A scientific literature search was performed, and the first search found 608 studies, and once the screening process was applied, 37 papers were included in this review. In the results, three research questions were answered that described the pests and diseases with high impact on avocado production, along with the data sources and the principal enabling technologies used in the identification of agroclimatic and phytosanitary events in avocados. Some challenges and trends in the parameterization of the technology in field conditions for data collection are also highlighted.

**Keywords:** avocado; internet of things; machine learning; deep learning; risk management; smart farming; pest and disease; climate conditions

## 1. Introduction

Avocado (*Persea americana*) is one of the most commercialized and profitable fruits in the international market [1,2]. This particular fruit is cultivated in nations characterized by tropical and subtropical weather conditions, many of them with emerging economies [3,4]. In 2021, the American continents produced 70.2% of the avocados marketed globally, where four Latin American countries are among the top five avocado producers in the world: Mexico (2.4 M tons), Colombia (980 K tons), Peru (777 K tons), and the Dominican Republic (634 K tons) [1]. Likewise, the significant increase in avocado production in these countries has been due to their entry and positioning in the global market, causing better opportunities for the producers of this fruit, accompanied by a boom in market demand for its functional compounds and benefits for human health, which represents a competitive opportunity for the region [5,6].

To achieve sustainable growth in agricultural production (e.g., avocado), one of the tasks of growers is to assess and quantify the risks to the farm in the event of a pest or disease outbreak in their crop, or damage caused by weather variations, to estimate the

impact and moderate the consequences on production or locate potentially damaged trees to respond on time and mitigate losses [7,8]. Establishing risk categories for agroclimatic and phytosanitary events (APEs) in crops makes it possible to define a differentiated treatment of crops according to the trees affected [9]. Nevertheless, despite the advancements in smart farming, a comprehensive framework has yet to be established to comprehend the specific contributions of crop-data-driven measures toward mitigating risks [10]. By utilizing crop data measurements to forecast the potential impacts of biotic and abiotic factors, the implementation of diverse measures during a single production phase can effectively diminish risks in various ways [11]. Therefore, the use of technology is key in agricultural production improvement strategies [12].

Recently, the fourth industrial revolution (4IR) has influenced a broad spectrum of industries and sectors through the introduction of new technologies, mainly in economic activities of the secondary sector (manufacturing, clothing, construction) and the tertiary sector (transportation, maintenance, marketing, communications), to reduce inefficiencies and improve market performance [13–20]. Similarly, the agricultural industry is experiencing a growing presence of digital transformation, attributed to the advancement and accessibility of information technologies (ITs) tailored to this domain. Nonetheless, agriculture remains among the least digitized economic sectors in the world, even in countries with fairly competitive agricultural systems and strong technological advances [21,22]. Today, ensuring people's food sustainability is very important, which makes agriculture one of the most relevant economic activities, leading to the transformation of traditional agricultural processes into techniques that are highly supported by technology, to increase crop production sustainably and increase economic opportunities for the families involved in these activities [23–25].

Emerging technologies, such as artificial intelligence (AI), specifically machine and deep learning (ML/DL), and the internet of things (IoT), are part of smart farming and today can be successfully employed to mitigate risks in production through the integrated use of ITs with predictive yield models [23,26]. Despite the availability of technologies, most of the field data needed for agricultural yield prediction are hosted by different organizations, and entities are reluctant to share data [26,27]. Similarly, very few technological applications and field databases on avocado crops in tropical areas are available, both those related to production and those related to the management of risks associated with APEs.

Some literature reviews have been conducted related to technology applications in the agricultural sector in general, but these studies do not address the specific management of pests, diseases, and climate changes in avocado crops using emerging technologies. Cravero et al. conducted three literature reviews associated with big data (BD) in agriculture [22,27,28]. The first review aimed to examine the advancements of technologies employed in BD architectures for agriculture, with a focus on climate change. The study highlights the tools utilized for processing, analyzing, and visualizing crop data, as well as the implementation of crop data architectures for effective monitoring of the weather, water, and soil conditions [28]. Following the same approach, the second review supplied insights into the utilization of BD and ML in the agricultural domain, without specifically addressing the adaptations and design of architectures for these systems to operate effectively in agricultural applications. It is noted that thanks to advancements in cloud technologies, data manipulation is no longer a complex process, although poor control of crop data and technical data visualization systems poorly understood by producers remain a challenge [22]. Additionally, the third review synthesized the available evidence about the current challenge of implementing ML techniques in agricultural BD applications. It noted the ML techniques and the main technologies used in agriculture and that it is imperative to express the necessity of modifying the current set of technologies by adapting ML techniques as the volume of data coming from farms increases [27]. Moreover, Toscano-Miranda et al. reviewed the detection and diagnosis of pests and diseases in cotton crops using AI techniques, finding several applications of techniques focused on image classification, image segmentation, and feature extraction from cotton production

data [29]. Likewise, Morella et al. reviewed the advantages present in the application of 4IR technologies in agricultural production, specifically in the supply chains of the agri-food sector, showing how emerging technologies improve the development of chains in this sector [30]. Also, this study identifies and examines the challenges involved in applying Industry 4.0 in agriculture and raises key indicators to help quantify the benefits of implementing the IoT, BD, and cyber–physical systems (CPSs) in this sector. Purcell and Neubauer conducted a review of the uses of digital twins in the agricultural sector [31]. The review aimed to identify recent trends and applications of this technology, to raise awareness and understanding of the potential and opportunities offered by digital twins in agricultural activities. Finally, Kountios et al. reviewed numerous current initiatives that use ITs to provide agricultural knowledge to producers, highlighting that knowledge of agricultural practices has been passed down from generation to generation through experiential learning and how technologies are widely used especially in the area of farmers' access to market knowledge, providing advisory services in agriculture and creating a competitive advantage in the sector [32].

Therefore, it is necessary to conduct a study related to avocado crops and the application of technologies in their production, to analyze the use of technological advances in the management of biotic and abiotic factors in avocado production farms, especially in tropical regions such as Colombia and the Andean region. Therefore, it is crucial to identify the factors associated with APEs in avocado crops [9,33]. For this reason, this paper is interested in finding the most relevant variables associated with APEs in avocado production, the incidence of these events in avocado crop productivity and their relationship with risk management, and the technologies used to detect and analyze pests and diseases in avocado. Consequently, primary studies conducted for the identification of pests, diseases, and climatic conditions that affect avocado crop yield were reviewed. Similarly, primary studies related to yield and risk management in avocado crops based on the monitoring of APEs were reviewed. Furthermore, primary studies on advances in the use of emerging technologies for data capture and analysis of avocado crop data were also reviewed in this paper.

Accordingly, this paper presents a systematic literature review (SLR) about the use of technologies for monitoring phytosanitary (pests, diseases) and agroclimatic (weather variables) agents in avocado crops. The objective is to show the state-of-the-art research on the management (prediction, detection, monitoring) of biotic and abiotic factors related to APEs in avocados, supported by applications of smart farming. The paper is structured in sections, where Section 1 is presented as the introduction. The methodology to perform this systematic review is presented in Section 2. The analyses and findings in context are presented in Section 3, followed by the discussion in Section 4. In the end, the conclusions of this review are presented in Section 5.

## 2. Materials and Methods

The methodology used in this paper focuses on the SLR. The PRISMA method [34] was used to select the research publications reviewed in this paper. Five stages of the PRISMA method were applied for the analysis of the reviewed papers: (i) eligibility criteria, (ii) information sources, (iii) search strategy, (iv) selection process, (v) preliminary analysis [35]. Additionally, the method proposed by Kitchenham et al. [36] was adapted to define the research questions (RQs) used in this SLR. The three research questions shown in Table 1 were proposed to achieve the goal of this review. The analysis started with this set of research questions, and query strings were constructed and used in the scientific databases selected, allowing to obtain relevant publications according to each research question. Each applied methodology step is outlined in the subsequent subsections.

**Table 1.** Research questions used in this SLR.

|  | **Research Question** |
|---|---|
| RQ1 | Which agroclimatic and phytosanitary events are involved in the development of pests and diseases in avocado crops? |
| RQ2 | How do agroclimatic and phytosanitary events affect productivity and parametric risk management in avocado crops? |
| RQ3 | Which sensors and advanced data analytics techniques are used for pest and disease detection and analysis in avocado crops? |

*2.1. Eligibility Criteria*

Table 2 presents the inclusion criteria (IC) and exclusion criteria (EC) established for this SLR. Since the highest avocado production area is Spanish-speaking and many studies associated with this product are concentrated in these countries, the search also included technological and scientific advances from the last six years published in English and Spanish.

**Table 2.** Inclusion and exclusion criteria.

| Criteria Type | Criteria Detail |
|---|---|
| Inclusion criteria | IC1 Research papers and conference proceedings related to APEs, crop yield, and technology use in avocado crops<br>IC2 Publications from 2017 to 2022<br>IC3 Publications in English and Spanish<br>IC4 Publications with full text accessible |
| Exclusion criteria | EC1 Papers that do not match the previous ICs<br>EC2 Books, chapters, reviews, editorials, abstracts, keynotes, and posters<br>EC3 Opinion pieces or position articles |

*2.2. Information Sources*

The data sources included several digital libraries (scientific databases) such as Web of Science for its scientific relevance, Scopus for its high scientific impact, and IEEE Xplore for its specificity in technological advances and computer science.

*2.3. Search Strategy*

The search strings were constructed as follows: (a) from the research questions, keywords for each string were obtained; (b) comparative analysis was applied to frame the results with research questions. Search strings were structured using the logical operators "AND" and "OR". The search string generated for each research question, together with the number of documents obtained for each query in the databases, are shown in Table 3. Each search string was utilized to search for relevant papers in the titles, keywords, and abstracts. Thus, 608 documents were initially obtained from queries to scientific databases.

**Table 3.** Search string used for each research question in this SLR.

| Question | Search String | Documents |
|---|---|---|
| RQ1 | (("climate change" OR "weather extremes" OR "plant disease" OR pest OR insect) AND avocado) | 487 |
| RQ2 | (("agricultural insurance" OR "crop insurance" OR "crop quality" OR "crop yield" OR "risk management") AND avocado) | 34 |
| RQ3 | (("deep learning" OR "machine learning" OR "artificial intelligence" OR "internet of things" OR "remote sensing") AND avocado) | 87 |

*2.4. Selection Process*

For the selection of the papers to be reviewed, the ICs and ECs previously presented in Table 2 were applied to the documents found in the scientific databases, with the help

of technological tools for the selection of publications [37]. In the first stage, papers were selected by evaluating their title and keywords to exclude papers unrelated to the research questions. Also, duplicate papers were removed. In the second stage, the abstracts of the preselected documents from the previous stage were examined. Thus, each paper underwent evaluation by applying ICs and ECs to determine whether it would proceed to the next step. During the third stage, the complete text of the candidate documents was reviewed for final selection, and the papers that not met the established criteria were excluded from the review. In the fourth stage, the studies obtained from other sources were included, especially papers identified through snowballing applied to the studies selected in previous stages and retrieved through Google Scholar and ScienceDirect. At this stage, the papers with the highest number of citations for each query were also added. These papers are the starting point for many of the studies reviewed, and for their inclusion, we omitted the IC2 that limits the time window for publication of the papers, obtaining two high-impact papers. This completed the total number of papers included in the review. Figure 1 shows the flowchart of the papers' selection process. Filters were applied to the selection process, where 35 papers were selected and 2 highly cited papers were added, for a total of 37 reviewed papers.

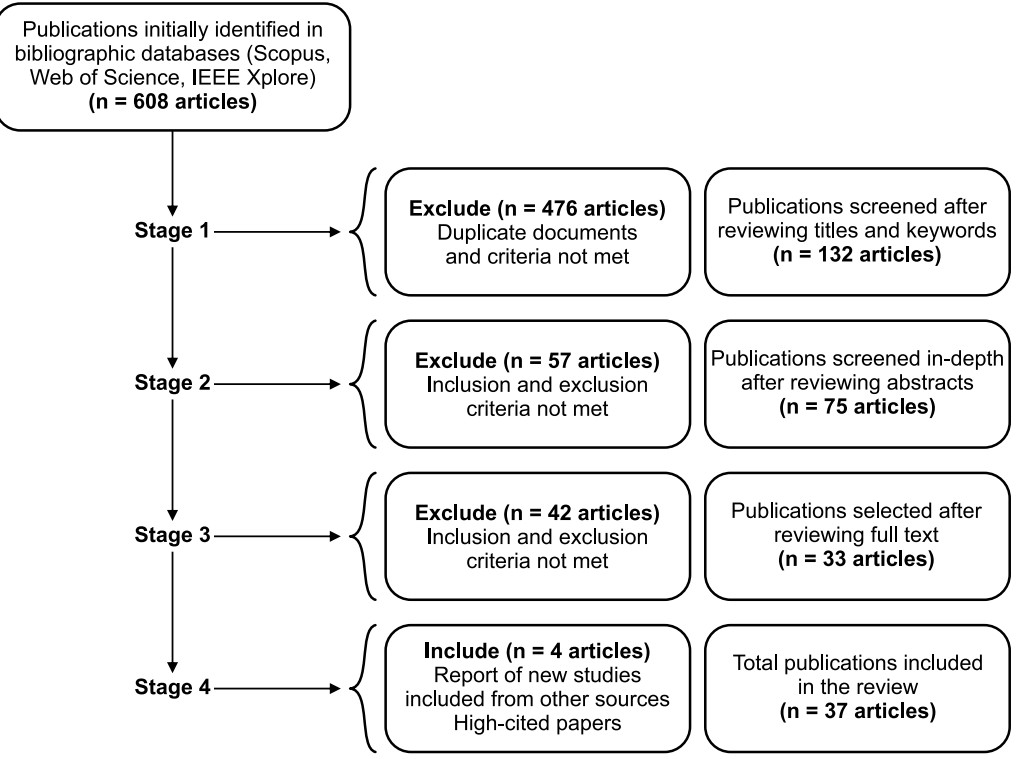

**Figure 1.** Stages followed in the selection process of the reviewed papers.

*2.5. Preliminary Analysis*

Based on the selection process, this SLR included a total of thirty-seven papers selected according to the eligibility criteria. The composition of the selected papers varies between publications in scientific journals and publications in conference proceedings. The journals with the highest number of selected articles are Computers and Electronics in Agriculture from Elsevier with four publications, and Agronomy from MDPI with three publications. These two publishers, together with Springer, account for about 57.14% of the selected studies, and the other publishers account for the remaining 42.86%. Figure 2 shows both the journals and the publishers with one or more publications selected for this SLR and related to the research questions. Thus, this figure shows the number of publications selected by each publisher in the first 35 selected papers.

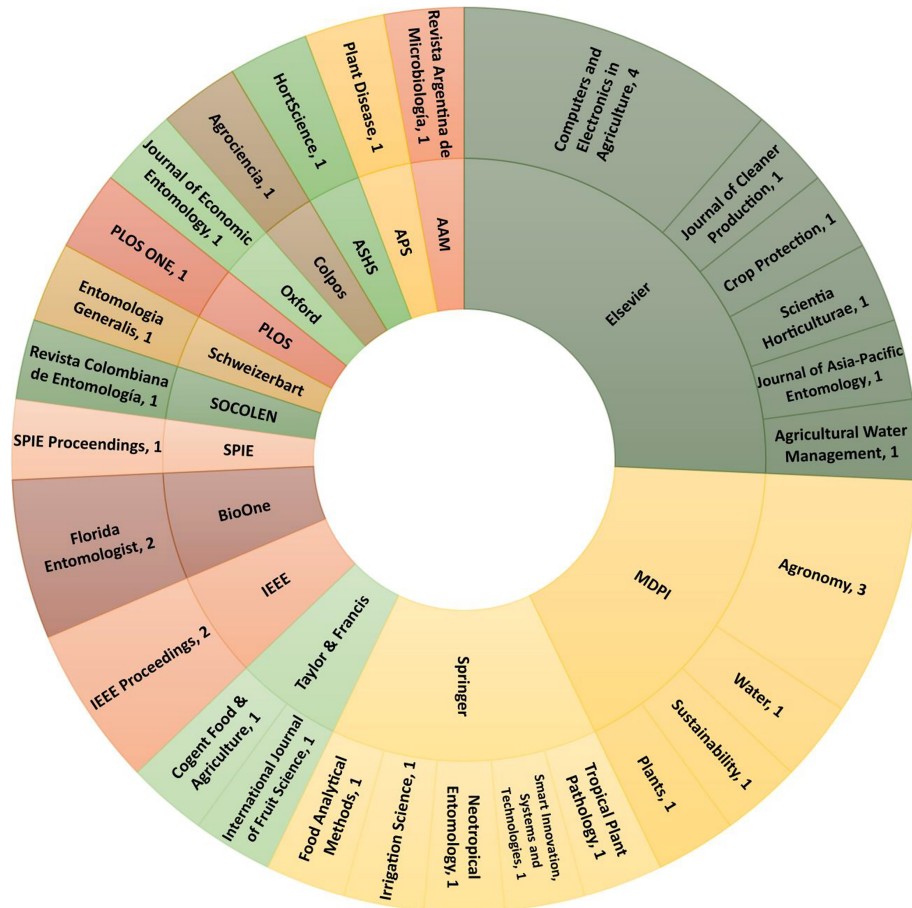

**Figure 2.** Publisher and journals names of the selected papers.

Analyzing the publication type, 33 (89.18%) of the selected publications correspond to research papers in scientific journals, and 4 (10.82%) publications are papers in scientific conference proceedings. Most of the first 35 selected papers were published between 2020 and 2022, representing 65.71%. The remaining group of these papers (34.29%) was published between 2017 and 2019. Figure 3 illustrates the yearly distribution of the first 35 included papers in this study.

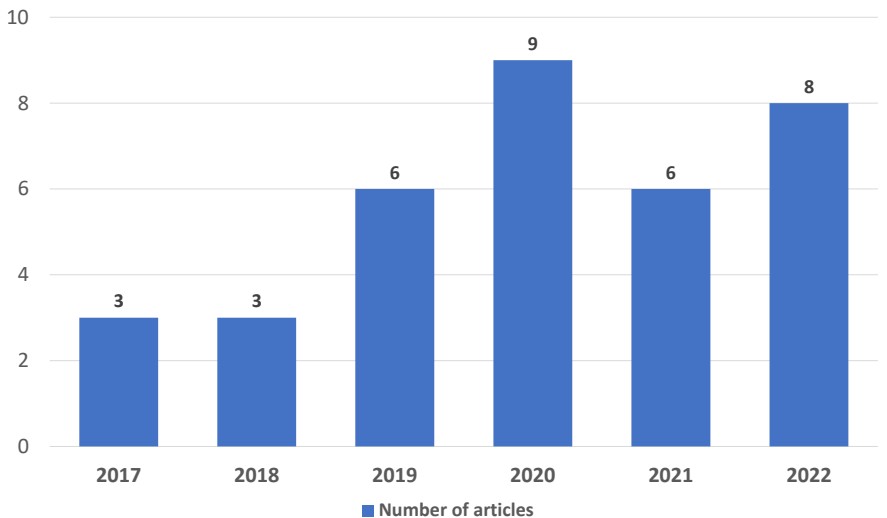

**Figure 3.** Number of publications per year in selected papers.

The distribution of the selected studies around the world is shown in Figure 4 and was conducted in 11 countries. The majority of the studies were from Colombia, with 9 papers (25.71%); followed by the United States, with 6 papers (17.14%); Mexico and Kenya, with 4 papers (11.43%) in each country; Peru and Israel, with 3 papers (8.57%) in each country; Spain; with 2 papers (5.71%); and closing with India, the Philippines, Switzerland, and Denmark, with 1 paper (2.86%) in each country.

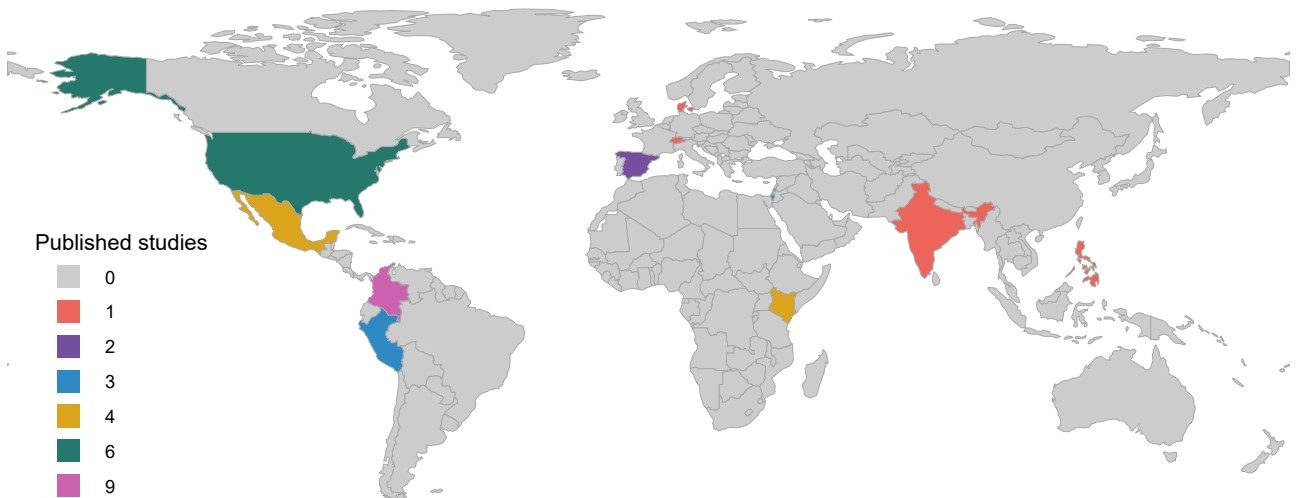

**Figure 4.** World map distribution of the reviewed research papers.

It is worth noting that Mexico, Colombia, and Peru, which are among the main global avocado producers, have implemented robust initiatives to enhance avocado production and quality. These initiatives include the development of tools for identifying insect pests that pose significant threats to crops. Furthermore, there is a strong emphasis on promoting comprehensive, cost-effective, and environmentally safe solutions for managing these pests [38,39]. Likewise, studies carried out in Switzerland and Denmark highlight the importance of avocados in the global market and promote the mitigation of risks caused by climate change, while the other countries mentioned above have promoted studies aimed at optimizing their avocado production under changing climatic and phytosanitary conditions [40].

### 2.6. Data Analysis

The sequential narrative was used as the primary method of data analysis for this SLR. In this method, specific subtopics of the selected studies are denoted by highlighting the main data points below the subtopics associated with the research questions. Additionally, the synopsis method was used to highlight important information points from the other pieces of information within the papers regarding the RQs.

## 3. Results

The analysis of the reviewed papers is presented below, with their main information. This information is organized according to the research questions posed and according to the focus of the papers reviewed.

### 3.1. Highest Cited Papers in RQs

The highest citing papers found for each RQ are an interesting starting point to understand the current trends in studies related to APEs in avocado crops. The results published in these papers serve as a starting point for several of the works analyzed in this paper.

In the case of RQ1, Ippolito and Nigro reviewed several studies on the use of biological control methods to manage postharvest diseases in fruits and vegetables, including

avocados [41]. The effectiveness of these methods is evaluated through laboratory and field experiments. The authors highlight the potential of biological control as a sustainable and environmentally friendly alternative to chemical fungicides. In addition, the negative effects of pesticide applications on non-pathogenic microorganisms and the resistance of biological control agents to fungicides are mentioned. This paper concludes by discussing the potential for genetic manipulation of biocontrol agents and the need for further research on preharvest applications of biocontrol agents for postharvest disease control.

In the case of RQ2, Lobell et al. analyzed the relationship between crop yields and climatic variables in California [42]. Multiple regression models were used to determine the most important weather variables for each crop, including avocado. The results showed that different crops had different relationships between weather and yields. The study also found that climate changes since 1980 had a mixed effect on crop yields, with oranges and avocados being the most affected. The authors suggest that these findings can be used to improve yield predictions and understand the potential impacts of future climate change on crop yields.

In the case of RQ3, Abdulridha et al. present a remote sensing technique to detect laurel wilt disease in avocado trees [43]. The technique uses multispectral imaging and two AI classification methods to differentiate between healthy trees, trees infected with laurel wilt disease, trees infected with Phytophthora root rot, and trees with nutrient deficiencies. The study found that the technique was able to successfully detect laurel wilt disease with 99% accuracy at the early stage. This automated technique could improve disease detection and management in avocado orchards.

### 3.2. Weather Variations and Effects in Avocados

In Erazo et al., the study indicates that water supply is a critical abiotic factor in the production of Hass avocado since it intervenes in the physiological processes that delimit both the period and intensity of the harvest, being directly related to the yield of avocado trees and the quality of the harvested fruit [44]. Weather events that cause a deficit of water in the crop decrease productivity potential, since rainfall is the main source of natural water for the crop, especially for tropical areas such as Colombia. Although the total annual sum of rainfall can supply the full amount of water required by an avocado crop, in periods when rainfall is very low and distributed heterogeneously over time, this source does not supply the amount of water needed by the crop. Therefore, an important finding in the modeling used to predict rainfall variations indicates that at least 99.8% of the avocado crops in these areas require assisted irrigation one month per year.

Alon et al. conducted a study researching the effects on parameters of avocado tree physiology resulting from the application of shading nets under extreme heat conditions [45]. Considering that the concurrent presence of high temperature and intense irradiance increases plant stress and harms plants [46], 60% shading screens on avocado trees significantly reduce leaf and canopy temperature by up to 4 °C during extreme heat events. Young Hass avocado plants have a heat damage threshold between 49 °C and 51 °C in low-light conditions [47], so the ability to shade netting to reduce air and leaf temperature can significantly reduce extreme heat damage and improve productivity.

The current global trend in climate change studies involves the application of modeling, risk analysis, and prediction techniques, as well as to search for alternatives that will enable agricultural systems to mitigate and adapt to climate variability in extreme situations, as stated by Ramirez-Gil et al. [48,49]. Consequently, a major challenge faced by avocado cultivation is to find alternative strategies for trees to adapt to the adverse effects of extreme variability in climatic conditions. A variety of strategies were evaluated to mitigate climate variability's adverse effects on avocado crops, as well as the impact that this variability can have on the occurrence of diseases such as avocado wilt. Similarly, the study highlights that various factors, including genetic characteristics, weather conditions, agronomic practices, harvesting standards, and the incidence of pests and diseases, influence avocado fruit quality in crop production. The main problems affecting Hass avocado quality were

identified at three stages (preharvest, harvest, and packing). One of the abiotic factors that can impact the quality of Hass avocados is the occurrence of reduced fruit size and necrotic seed. This phenomenon appears to be linked to periods of water deficit during fruit filling, which can be influenced by the "El Niño" climatic phenomenon [48]. Among the effects of biotic factors is the alteration of the epidermis, malformations, and the fruit pulp, caused by arthropods such as Melolonthidae complex (*Astaena pygidialis* Kirsch), thrips (Thysanoptera), the bug monalonion (*Monalonion velezangeli* Carvalho and Costa), and scales (Hemiptera: Coccoidea) [49].

### 3.3. Pests and Diseases Incidence in Avocados

Regarding pests affecting avocados, López Galé et al. identified 42 species of phytophagous insects in avocado crops located in 78 farms in mountainous regions of the Colombian Caribbean [50]. Light traps were used to capture insects such as moths, "marceño" beetles (Melolonthidae), and branch borers (Coleoptera: Curculionidae), which are registered as pests in avocados, and manual capture was used for other insect species. According to their taxonomic classification, the most frequent insect groups encountered in the area were branch borers, scale insects (Hemiptera: Coccomorpha), and termites (Blattodea: Isoptera). These pests with high frequency in crops; in particular, branch borers are considered a group of insects that cause damage of economic importance in avocado crops because of their difficult control, the spread of diseases, and the death of trees.

Holguin and Mira selected commercial avocado crops in Antioquia, the department with the highest production of this fruit in Colombia, to determine the presence of adult beetles affecting small fruits and young leaves [51]. Using ultraviolet light traps installed in the crops and direct inspection of avocado trees, nine beetle species were captured. Among the insect species captured, *Astaena pygidialis* Kirsch was the only insect species with an incidence at all sites where samples were collected for the study, while the remaining eight insect species were only found at one or a few sampling sites. Additionally, *A. pygidialis* Kirsch is the only beetle detected that caused damage to the epidermis of the fruit and skeletonization of the leaves, finding reports of this same damage in other regions of Colombia, affecting the productivity of the crop and the commercialization of the fruit due to its damage.

In Valencia Arias et al., the authors observed the specific behavioral patterns between Melolonthidae complex beetles and avocado crops to estimate their economic effect on the production of this fruit in Antioquia, Colombia [52]. In this study, adult beetles were collected in the rainy months of the year (March to May) using light traps for three years, and the observations revealed that young leaves and shoots damaged by the pest had the potential to become necrotic following defoliation. Additionally, they found that fruit infections caused by these beetles typically manifested as superficial scarring that increased in size as the fruit developed. The distance radius of damage caused by the pest is 15–30 m, which indicates the potential capacity of a beetle to attack a group of up to four avocado trees in the surrounding area [52]. Due to the aesthetic impact caused by these beetles, affected fruit are often labeled as "industrial" avocados, a category considered to be of lower quality. As a result, these industrial fruits are sold in the local manufacturing market at a lower price.

The interactions between the physico-chemical environment, pollinator insect (*Apis melifera*) introduction, and pest management in avocado crops were evaluated by Toukem et al. [53]. This approach is being investigated as a potential solution for achieving sustainable fruit production while minimizing negative impacts on the environment. Using generalized linear mixed models, this study found significant interactions between vegetation productivity, pollinator introduction, and pest management in a set of avocado fruit, decreasing the presence of avocado pests and increasing fruit weight by 6% compared to crops without integrated pest management. Furthermore, another previous study examined the correlation between vegetation productivity and the abundance and distribution of crucial avocado pests on small-scale farms in Kenya, expecting a correlation of the

normalized difference vegetation index (NDVI), rainfall, and temperature of the area [40]. Thus, the aim is to understand the special occurrence of pests and develop cost-efficient and sustainable pest management practices to increase avocado crop productivity.

### 3.3.1. Relationship between Weather and Pests

Climatic variations can directly or inversely affect the proliferation of crop pests. Luna et al. contributed to developing pest risk assessments for fruit growing by determining the geographic distribution of five important avocado insect pests located in Mexico (*Conotrachelus aguacatae, Conotrachelus perseae, Heilipus lauri, Copturus aguacatae, Stenoma catenifer*) and how these pests interact with commercial avocado growing areas [54]. To accomplish this, the study employed the maximum entropy method (MaxEnt) [55] to evaluate the impact of bioclimatic factors on the propagation of these pests. The model incorporated 19 global climate variables and elevation data to analyze their influence. Altitude, temperature regularity, and rainfall seasonality emerged as the key variables that exerted the strongest influence on the potential propagation of the pests observed, contributing 72.6% for *C. aguacatae*, 76.1% for *C. perseae*, 79.1% for *Copturus aguacatae*, 77.0% for *H. lauri*, and 66.7% for *S. catenifer*.

In Velázquez-Martínez et al., the authors monitored the population fluctuations of *S. catenifer* Walsingham pests in Hass avocado crops in Mexico, exploring their relationship with relative humidity and temperature [56]. For this study, pheromone-baited traps were used in four avocado plantations. The infestation percentages recorded were 65% for the first orchard, 55% for the second orchard, 17% for the third orchard, and 12% for the fourth orchard. A negative correlation was found between the number of *S. catenifer* captured, relative humidity (r = −0.21), and temperature (r = −0.25). The insect captures were recorded within a temperature range of 15.8 °C to 25.4 °C, with relative humidity fluctuating between 41.1% and a maximum of 96.1%. Consequently, the estimated relative risk of the two bivariate distributions (temperature/relative humidity) suggests that the risk is nearly negligible across the entire temperature range when relative humidity is low. Notably, during low temperatures, relative humidity emerged as the variable with the most significant impact on the risk of being affected by this particular pest.

Ibrahim et al. used total rainfall, average temperature, and relative humidity, together with data from pest counts in traps, to model population dynamics of two avocado pests (*Bactocera dorsalis* and *Ceratitis* spp.) in Kenya [57]. The study integrated data fuzzification techniques to develop fuzzy neural network (FNN) models. These models were utilized to predict pest counts, aiding in decision-making and determining the optimal timing for interventions within integrated pest management strategies. The FNN models obtained a coefficient of determination ($R^2$) greater than 0.85 for results in predicting pest dynamics in avocados. These models can be used for pest identification including predicting the levels of damage (low, medium, high) caused by pests on crops. Similarly, in the study conducted by Odanga et al., the authors analyzed the correlation between weather data (temperature and precipitation) and their influence on the spatial distribution of *Thaumatotibia leucotreta* and *B. dorsalis* and the infestation of these two pests of avocado trees in Kenya [58]. With 12 months of data collected on the crop, the temperature was found to strongly influence the seasonal population growth of *B. dorsalis* (r = 0.94), but not *T. leucotreta* (r = −0.15), in the dry season. The *B. dorsalis* pest exhibited its highest abundance during the warmer dry period immediately following the short rainy season, while its population density was comparatively lower during the long rainy season. On the other hand, the *T. leucotreta* pest demonstrated a presence throughout the year, indicating its ability to develop in any weather season.

### 3.3.2. Relationship between Weather and Diseases

Weather variations also have an impact on the presence of diseases in avocado crops. Thus, estimating the spread of diseases in the crop can be useful to mitigate the damage caused by these diseases. Burbano-Figueroa et al. estimated the potential dispersion of

avocado wilt and branch dieback in crops of this agricultural product in the Serranía del Perijá (Colombian–Venezuelan border) [59]. This disease is attributed to the fungus *Bionectria pseudochroleuca* and is dispersed by ambrosia beetles. It leads to severe detrimental effects on the crop. The distribution model was approximated using the correlated MaxEnt method, which is an ML algorithm commonly employed to forecast the likelihood of spatial distributions based on presence records [55]. The model incorporated pest field variables along with 20 climate variables derived from historical weather data. The findings indicate that the disease occurrence was more prominent in areas characterized by low rainfall during the wettest month, typically around 200 mm, and a complete absence of rainfall in the driest quarter. Conversely, the occurrence of the disease exhibits a sharp decrease as rainfall levels increase, both in the wettest month and in the driest quarter. In addition, disease occurrence is absent in areas where the wettest month and the driest quarter experience rainfall levels above 600 mm. Furthermore, the disease occurrence is absent in areas where the wettest month and in the driest quarter experience rainfall levels exceeding 600 mm.

Menocal et al. evaluated the interaction of flights by ambrosia beetles, the vector of laurel wilt, with avocado host trees [60]. The study found these beetles flying at heights of up to six meters corresponding to all parts of avocado trees, where the captured spices could have vectored the pathogen causing laurel wilt between crops. In addition, pests were observed to initiate their flight before sunset ($\sim$1 h earlier), showing that temperature is a critical abiotic factor for beetle flight activity (negative correlation). This helps to understand the distribution dynamics of the insects (from biological understanding) and their potential role as disease transmitters in avocado crops.

In Tapia Rodriguez et al., the authors proposed to determine peaks of anthracnose infestation using geostatistical methods in avocados, to elaborate disease distribution maps, and to determine the infested area [61]. The aim was to create integrated management strategies that favor the environment, in addition to the development of efficient sampling plans that will help in making decisions on disease control in the crop. Additionally, a comprehensive definition of the management strategies for anthracnose disease in avocado crops was provided by Kimaru et al. in their study [62]. Avocado growers worldwide need to control anthracnose disease to ensure that avocado fruits are of high quality. Farmers primarily employ pruning as a cultural method to mitigate anthracnose disease in avocado trees. Pruning helps improve air-flow and reduce moisture within the leaf canopy, creating an unfavorable environment for the fungal growth responsible for anthracnose. Furthermore, chemical control methods involving the use of fungicides are implemented to mitigate the incidence of anthracnose disease in avocados. Nevertheless, the widespread use of pest chemical control has caused global preoccupation due to the potential adverse effects of pesticide residues on human health, animal welfare, and the environment.

*3.4. Risk Management and Yield in Avocado Crops*

Reints et al. studied how Hass avocado growers in California adopt irrigation practices and technologies to sustain their crops in times of water scarcity and high salinity, using an analytical model fitted with an empirical approach [63]. With a typical average water consumption of approximately 12,200 m$^3$/ha per year, the cost of irrigation water significantly impacts the profitability of crop cultivation. Typically, water utilities charge prices for water ranging from USD 1200 to 1300 per acre-foot. The results indicate that the 123 surveyed growers employ various technologies to sustain or enhance profitability in their Hass avocado crops while managing the risks associated with water availability fluctuations. Growers utilize several effective water management technologies and practices to optimize water usage. These include the use of soil moisture measurement devices, irrigation calculators, drip irrigation systems, and tree management techniques aimed at reducing water consumption, among other strategies. Regional climates and water conditions play a crucial role in shaping farming practices. In regions characterized by harsh climates or limited water availability, farmers are more inclined to adopt efficient water

management technologies and practices. This adoption is driven by the aim to minimize production costs and optimize water usage in challenging environments.

The tolerance of Hass avocado to water salinity was explored by Acosta-Rangel et al., given the low quality of water used in agricultural production, which has a lower availability and quality due to drought and extreme temperatures [64]. As avocado is extremely sensitive to salt, identifying the effects of salinity on plants allows actions to be taken to ensure the sustainability of a crop. The study revealed that irrigating Hass avocados with high salinity water resulted in a 44% increase in canopy damage and a 50% reduction in tree survival. The control group of trees produced an average of 5.9 to 7.3 kg of fruit per tree, while the trees subjected to the salinity treatment yielded only 1.5 to 2.8 kg of fruit per tree. This salinity treatment caused a significant yield loss of over 68% compared to the control group. Despite irrigation with low-quality water not causing water stress, the salinity treatment caused leaves to burn visibly, resulting in poor carbon assimilation and poor yield.

Silber et al. evaluated the water demand required by Hass avocado for the growth of the fruit and identified the consequences of water deficit on crop yield during the phenological phases [65]. Applying five treatments in the experimental design (no water stress, excessive irrigation, regulated deficit irrigation, no irrigation or fertilization, and constant water stress), it was found that trees treated without water stress obtained a high yield (between 25 and 31 avocados tons/ha), while trees subjected to water stress had a significantly lower yield (between 16 and 21 avocados tons/ha). Moreover, irrigation plays a crucial role in facilitating tree growth by providing water. It is important to note that irrigation management also impacts nutrient availability in the soil. The interrelationship between water and nutrients must be carefully considered and managed to ensure optimal plant growth and development. Therefore, regulated irrigation management to cope with the water deficit in the Hass avocado crop is necessary during the main fruit-growing season.

In Moreno-Ortega et al., the authors conducted a study to assess the adequacy of the current water supply to meet the water needs of Hass avocado crops in Spain, specifically in the subtropical region of the country [66]. The aim was to determine the extent to which water usage can be enhanced for improved crop performance in the area. This study examined the productivity of water in a Hass avocado crop during six consecutive seasons as well as evaluated both the physiological and agronomic effects in the trees when applying five treatments during two consecutive seasons. The findings revealed that avocado yield and fruit quality were significantly influenced by irrigation levels above or below the range of 192–240 L/day per tree. The average yield of fruits in the entire crop, when subjected to traditional irrigation practices within the range of 160–200 L/day, was determined to be 10.34 tons/ha during the period from 2012 to 2018. Within this average, the on-crop season yielded 16.5 tons/ha, while the off-crop season yielded 8.7 tons/ha. These findings highlight the importance of proper irrigation management in achieving optimal avocado production and maintaining consistent fruit quality.

Grüter et al. analyzed the current and future suitability of three crops of interest (coffee, cashew, avocado) globally, taking into account the climatic and soil requirements of each crop, and identified and discussed global and regional trends [67]. This study used a decision support system based on geographic information systems, supported by a multicriteria evaluation of climatic and biophysical parameters (soil texture, pH, salinity, temperature, precipitation, humidity, among others), to model the present and future suitability of the crops of interest. The findings indicated that the current suitability of avocado production in the top four producer countries is constrained by low and high rainfall as well as low temperatures in the cooler season of the year. In the future, the suitability of avocado production is expected to be primarily influenced by changes in rainfall patterns, with both wetter and drier conditions having positive or negative effects. Additionally, there will be a lesser impact from the increase in minimum temperature during the coldest month, which is associated with positive changes in suitability. Temperature and precipitation changes are expected in Central America, West Africa, and Southeast

Asia to have both positive and negative impacts in these regions. By contrast, the highly suitable areas to grow avocado crops will decrease globally to 41% by the representative concentration pathway for the future climate conditions in the year 2050.

Caro et al. reported that from 2000 to 2016, international trade progressively influenced the amount of water used to produce avocados [68]. They found that almost one-third of the water use associated with avocado production is due to international demand and related trade. Higher values of the water stress index indicate more competition among agricultural activities. Since all the largest exporters present values oscillating between 3 and 4, they all fit within the category of high water stress (40–80%). In particular, Mexico (3.84) and Chile (3.57) score the highest among avocado exporters in terms of water stress index, whereas the main importers have generally lower scores. The overexploitation of water resources driven by the avocado trade can have detrimental effects on the environment, particularly in economically disadvantaged countries where avocado exports contribute significantly to economic growth. This overexploitation may exacerbate environmental challenges and have negative consequences for water availability and quality in these regions.

The light response of two avocado cultivars to sporadic frost stress can have negative results on crop yield and fruit quality, as stated in the study conducted by Weil et al. [69]. The focus of the study was on the photosynthetic parameters of these avocado varieties, and the researchers examined their responses to frost stress in four instances both in the laboratory and in the field conditions. The study found that before frost stress, approximately 40% of the total energy absorbed in Hass avocado was allocated to photoprotection mechanisms in the leaves, 30% was used for photochemistry, and the remaining energy was passively dissipated. However, after six hours of frost stress in lightless conditions, the energy allocated to photochemistry was significantly reduced, indicating a reduction in active energy utilization. This reduction in energy allocation to photochemistry could be a response to minimize potential damage and conserve resources in the face of stress. It was also noted that under these conditions, tissue dehydration may occur, potentially leading to water limitation within the avocado tree. This highlights the complex interactions between frost stress, photosynthetic processes, and water availability, which can impact the overall health and performance of avocado plants.

Campisi-Pinto et al. aimed to identify the nutrient concentrations in Hass avocado tissue associated with high fruit yields specifically those exceeding 40 kg per tree [70]. They provided insights into nutrient management practices for achieving and surpassing the targeted yield by establishing optimal nutrient levels for predicting yields above this threshold. The study found strong correlations between certain nutrient concentrations in avocado tissue and yields above 40 kg per tree. The findings suggest that high nutrient concentrations due to current fertilization practices may be causing nutrient imbalances, which limit productivity. Additionally, addressing nutritional problems can increase crop yield, fruit size, and fruit quality before these problems alter flower retention and fruit set.

In Ramirez-Gil et al., they conducted a study on the economic impacts of wilt disease in avocados at different stages of crop development in Antioquia, Colombia [71]. According to the researchers, plant mortality, fruit quality losses, and plant mortality had a total economic impact. The study revealed that in the northern, eastern, and southwestern zones of the department, the total economic impacts amounted to USD 635, 637, and 678 per hectare, respectively. These findings highlight the significant financial losses incurred due to wilt disease in avocado crops within the study area. The observed heterogeneity in the production systems of the studied regions results in varying production costs, which stem from the utilization of different practices and inputs, as well as the existence of different selling prices for the fruit. Consequently, the economic impacts of wilt disease in avocados were assessed based on the direct effects of tree deaths and reduced vegetation production, as well as the additional costs derived from the replacement of affected trees and disease management in general. This comprehensive approach considers both the immediate losses incurred from plant deaths and reduced yields, as well as the long-term costs related to mitigating the disease and reestablishing productive avocado trees.

*3.5. Emerging Technologies in Agriculture Applications*

The application of Industry 4.0 in the agricultural sector is often referred to as Agriculture 4.0. This term is simply mentioned to refer to the use of these digital techniques within precision agriculture [72]. However, sectoral variances between industry and agriculture must be taken into consideration when estimating the impact of emerging 4IR technologies in agricultural applications, such as avocado production. Through Industry 4.0 enabling technologies, it is possible to increase the precision agriculture solutions available to achieve similar benefits to the industrial sector and exploit new business opportunities for producers [73]. Benefits include digital individualization, demand orientation, sustainability, automated knowledge and learning, and productivity optimization [72]. Thus, the use of enabling technologies such as CPSs, Industrial IoT (IIoT) components, cloud computing, unmanned aerial vehicles (UAVs), BD, and ML in agriculture represents an opportunity to achieve efficient irrigation solutions, pest management, and pesticide control in crops and weather and soil condition monitoring, among others, highlighting the requirement to adapt these technologies to the basic structure of agriculture [72,73]. For this, Agriculture 4.0 requires the establishment of technological standards to ensure the interoperability of enabling technologies used in agricultural applications. The main issue in this field is the obligation to share data and communication standards that connect all systems covering all areas of farming, in addition to ensuring that technological equipment remains compatible with new developments and is supported over time by manufacturers and other industries, given the longevity of agricultural equipment [74].

With their advances, IoT technologies play an important role in various agricultural applications, thanks to the capabilities they offer, including basic communication infrastructure and a variety of services, such as in situ and remote data acquisition, information analysis based on cloud-hosted applications, decision support, and automation of agricultural processes [21]. Currently, the major applications of IoT in smart farming are oriented to water and nutrient monitoring, pest and disease monitoring, soil monitoring, crop health monitoring, environmental monitoring, and machinery control [21]. On the other hand, ML algorithms allow for optimizing task performance by generating efficient relationships between previous data input and outputs obtained from previous experiences, so the more data used, the better the algorithm will perform. As a central result, the ability of the ML algorithm to deliver correct predictions will rely on the set of rules learned from previous exposure to data of equal similarity [75]. Recently, applications of ML algorithms in smart farming are very common to detect and monitor pests and diseases automatically, finding also significant use of DL algorithms for these tasks [76]. A trend is focused on the use of specific algorithms based on DL for the detection of diseases in plant leaves [76]. However, this becomes a limitation in avocado crops since many pests and diseases take time to manifest themselves through the leaves of the plant.

*3.6. Data Collection Techniques Applied in Avocado Crops*

In situ and remote sensing technologies are used to detect damages caused by pests and diseases in avocados. Sensing in situ techniques include direct observation of crops and the collection of plant samples for subsequent analysis in the laboratory. Concerning remote sensing, the techniques use technologies such as drones and satellites to capture images and data of crops from a distance. Both methods have advantages and disadvantages but complement each other to provide a more complete assessment of avocado crop management. With the data obtained from avocado crops through these technologies, it is possible to apply advanced analytical techniques for processing and analysis, obtaining in some cases estimates of the incidence and damage caused by biotic factors as early warning.

3.6.1. Data Collection with In Situ Technologies

Applying developments focused on the IoT, Ramirez-Gil et al. designed a low-cost electronic prototype to collect and transmit weather-related data, including soil moisture and temperature, in avocado crops [77]. Furthermore, as part of their study, researchers de-

veloped a mobile application specifically designed for real-time reporting of early warnings associated with the Hass avocado wilt complex. Using the Arduino prototyping platform, the IoT system consisted of the power supply, sensors to measure weather variables, and data processing and storage in the cloud. The set of components used in the prototype cost approximately USD 1000, denoting its low cost compared to commercially available weather stations for agricultural applications. The device, installed in an avocado crop located in eastern Antioquia, had a data validation of 97.8%, 98.3%, 97.5%, and 97.7% for precipitation, sunshine, relative humidity, and temperature values, respectively, indicating the authors the reliability and low variability of data collection by the IoT device from these metrics.

Mejia Cabrera et al. designed a semi-automatic method for the detection of the *Lasiodiplodia theobromae* fungus in the Hass avocado production zone in Peru, which is characterized by producing a canker around the stems and branches of the tree [78]. For this purpose, using a semi-assisted system with a 12.1 Mpx Canon SX50 HS camera, the authors collected 150 images in situ (uncontrolled environment), composed of 60 healthy branches and 90 diseased branches with the canker, at a distance of 20 cm concerning each branch. The configuration was established so that the illumination was adequate, avoiding histogram saturation and obtaining photographs with good performance for processing in the other part of the system, consisting of a mid-range laptop with a model software developed in Python. Thus, the authors propose a step for the development of techniques that allow the automatic analysis of data from avocado cultivation.

Rivadeneyra Bustamante and Huamán Bustamante proposed a method to collect thermal and multispectral images of leaves in Hass avocado crops, oriented towards the anomalies detection in the avocados using computer vision techniques [79]. For this purpose, 499 thermal images and 499 RGB images of Hass avocado leaves were captured in situ, selecting 100 images of each type for analysis, considering their visual and thermal particularities to be useful for anomaly detection. The images were taken with a FLUKE TiS 45 infrared camera, while the ambient temperature was measured with a FLUKE 80bk-a thermocouple to contrast this variable with the values recorded in the thermal images. From the collected images, the authors proposed extracting their characteristics to classify avocado leaves as healthy and those affected by pests or diseases.

Some limitations to the data collected in the studies analyzed in this subsection derive from the practical considerations to be taken into account in the use of the prototypes. In [77], the stationary location of the prototype in the crop can have considerable variations concerning the microclimate of each avocado tree, particularly the relative humidity, which has a high incidence in the emergence of wilt. In [78], the frequency of monitoring the crop to collect images can lead to a wide bias in favor of the advance of canker in avocado trees, adding that a more significant number of images should be acquired to include other types of diseases as well. In [79], overexposure of the leaves to solar radiation at the time of capturing the thermal image can cause considerable variations between the temperature recorded on the leaf and the ambient temperature, causing possible limitations in the interpretation of the results. Additionally, in these papers, analytical techniques were applied to the data collected from avocado crops, which are reviewed subsequently in the Section 3.7.

### 3.6.2. Data Collection with Remote Sensing

Abdulridha et al. presented a non-destructive method for the identification of avocado trees infected with laurel wilt in the early-stage and the late-stage, and differentiate this disease from trees affected by other causes with similar symptoms (i.e., nitrogen and iron deficiencies) [80]. Through remote sensing techniques, the study used a portable system for data collection composed of a multispectral camera configured in the visible-near infrared spectrum (400–970 nm). To capture images of the group of plants used as controls, they obtained healthy leaves from 1-m-tall avocado plants grown in pots under outdoor conditions. The researchers included four treatment classes (laurel wilt, iron, nitrogen,

and healthy). For disease, they randomly selected ten plants and inoculated the pathogen. For each treatment, five scans per leaf were taken using an SVC HR-1024 spectrometer under laboratory conditions. These data were averaged every 10 nm and 40 nm given the availability of commercial filters for prototyping a cost-effective sensor. From the collected data, 23 vegetation indices were calculated to differentiate the four treatment classes.

Based on his previous work, Abdulridha et al. presented and evaluated subsequently an automated analytical technique that early detects diseases in avocado trees using remote sensing to detect Phytophthora root rot, laurel wilt, nitrogen, and iron deficiency, and differentiate healthy from affected trees [43]. For this study, ten avocado plants with the disease were randomly selected, while five healthy avocado plants were used as a control group. The authors evaluated two systems for capturing images of the group of avocado trees grown in pots under outdoor conditions. The first sensing system used a Tetracam Tetra mini multispectral camera with six independent optical sensors of 1.3 Mpx each and user-parameterizable bandpass filters. The second sensing system used a modified Canon SX260 NDVI camera with a 12.1 Mpx optical sensor and a 32 mm filter added on the front rails of the camera. The images were taken from a platform located above the trees 10 m above the ground in sunny conditions. In both systems, images were stored locally on a removable memory card in each camera for subsequent analysis. This study used two methods to capture the data. The first covered the entire green area of the canopy of each plant, while the second covered the small polygons corresponding to the leaves of each plant, randomly selecting eight leaves per plant.

Pérez-Bueno et al. examined remotely sensed images (spatially resolved camera video, multispectral images, long-wavelength thermal images) to identify the presence of white root rot in avocados [81]. Data were collected in a one-hectare avocado crop, where 24 trees were selected to estimate their aerial symptoms at four levels (healthy, medium wilt, severe wilt, and death). The sensing system consisted of a DJI S900 UAV piloted by remote control, on which three cameras were mounted: a GoPro Hero-3 video camera, an ADC Micro multispectral reflectance camera, and an Optris PI-450 long-wavelength thermal camera. The multispectral camera was equipped with three filter bands (560 nm, 660 nm, 830 nm) in the green, red, and near-infrared spectrum regions, respectively. The researchers conducted eight independent flights to collect the crop data, which were stored on a system-mounted Optris PI LightWeight miniature computer. With the data, the NVDI of each tree was calculated to locate its state in one of the four levels of affectation.

A proposal to analyze the variables associated with water stress in avocado crops by acquiring multispectral images of the avocado crop using a camera installed on a UAV was presented by Castillo-Guevara et al. [82]. The system consisted of a DJI Matrice 600 Pro drone and a Parrot Sequoia 4-band multispectral camera (green, red, red-edge, near-infrared). At a height of 70 m above the ground, captures were taken over a 634.22 m$^2$ avocado crop area. For data collection, the trees were not irrigated for four weeks to cause water stress in the studied crop area, capturing one image per week. Setting 12 threshold values using data from week 0, the images captured in two bands (red and near-infrared) were segmented using the Otsu multithresholding technique, and the NDVI of the crop was obtained for the data of the following weeks. With this, the study found that the avocado tree manifests its water stress in a concentrated form from the edges towards the stem, which promotes the creation of tools to help growers make decisions to improve irrigation and differentiated treatments in their avocado crops.

Several limitations to data collection in the articles analyzed in this subsection arise from the conditions of the environment in which they were collected. In [43,80], although the plants were located in outdoor conditions since they were not planted in the soil or areas with typical temperatures and altitude for avocado growth, they may generate behaviors of disease progression different from those observed in avocado crops. In [81,82], although the data were collected in the field, wherein the first case the diseased trees were previously identified, while in the second case, the trees were subjected to water stress, some factors on the control group could not be controlled (dispersion of the disease,

rainfall), which may generate a bias among the data obtained. The analytical techniques that were used to process the avocado data collected in these studies are further reviewed in the Section 3.7.

*3.7. Analytical Techniques Used to Process Avocado Crop Data*

Once data are collected on avocado crops, applying analytical techniques is key to detecting and estimating damage caused by pests and diseases. Early detection of these factors allows minimizing the risks that can decrease yield and quality of production in the crop, through an effective management of their causes. ML and DL algorithms are an opportunity to process and analyze these data.

Ramirez-Gil et al. used image processing techniques and multivariate time series analysis to analyze the data collected by the IoT prototype and the mobile app [77]. Specifically, they used vector autoregressive modeling (VAR) to diagnose avocado wilt causal agents based on identifiable patterns in the avocado crop data. To facilitate the implementation and functionality of these techniques in the mobile application, a pretrained neural network classifier was integrated into the cloud-based IoT data storage platform. This innovative approach enables timely detection and effective management of the avocado wilt complex by providing growers with a tool for early warning, accurate diagnosis, and proactive risk management. The developed prototype of the mobile application demonstrated a strong correlation of over 90% when compared to data obtained from traditional weather stations. Additionally, the early warning system integrated into the application achieved a prediction accuracy of over 70% for the variables related to the avocado wilt complex.

Mejia Cabrera et al. used a convolutional neural network (CNN) as a classifier in combination with image processing techniques to detect the canker in the images taken from the avocado branches in the crop [78]. Among the image processing techniques used by the authors is the resizing of the photographs, followed by the graph-cut technique to extract the region of interest. Thus, the preprocessed images were used as input to the CNN classifier trained with the 150 images taken at 10 epochs. However, due to the small number of images available, the neural network constructed by the authors was a shallow CNN. The tool determines the segmented area corresponding to the fungus infection, obtaining a 93% accuracy in the positive classification of the disease presence.

Rivadeneyra Bustamante and Huamán Bustamante proposed a method with the steps to extract the characteristics of the photographs taken of avocado leaves in the crop [79]. This allows the avocado leaves to be classified as healthy and those affected by a pest or deficiency. The method uses the K-Means algorithm for the segmentation of the images and a support vector machine (SVM) model for the classification of leaves into healthy and diseased. The 100 images of avocado leaves taken in situ by the researchers consisted of 20 photographs of healthy leaves and 80 photographs of diseased leaves. The k-means algorithm was composed of 15 clusters that delivered 30 descriptors of the avocado leaf thermal image. Together with the calculation of the normalized difference green-red index of the RGB images, the obtained descriptors were inserted into the SVM algorithm with a 70-15-15 distribution for training, validation, and testing. Thus, the classifier obtained an accuracy of 82.67% in the detection of anomalous visual characteristics in avocado leaves.

Abdulridha et al. employed the methods of decision trees and multilayer perceptron (MLP) neural networks for image classification [80]. MLP emerges as a popular classification technique renowned for its ability to learn through examples and generate a tailored function with parameters including weight, bias, and network topology, and has found extensive application in remote sensing classification within numerous agronomic investigations [38]. Using the averaged 10 nm and 40 nm reflectance bands in the multispectral images of avocado leaves, the MLP classifier was able to distinguish between the four classes (laurel wilt, iron, nitrogen, and healthy) at both the early and late stages of disease progression. This classifier obtained results in a range of 98–100% positive classification percentages for all data sets, while the decision trees had a percentage of 82%. Thus, they found that the use of MLP provides high accuracy in detecting trees infected with early-stage

laurel wilt and differentiating other biotic and abiotic factors. Similarly, in [43], the steps employed were acquisition, preprocessing, segmentation, feature extraction, and classification of the multispectral crop images. Using a neural network applying the MLP method, together with the K-nearest neighborhood (KNN) as a classifier method in combination with the analysis of the previously preprocessed images, the automated method was able to successfully discover early-stage laurel wilt, showing that a low-cost remote sensing method can be used to make a distinction of healthy plants from those not affected by this disease. In all experimental scenarios, the implementation of Tetracam (with 6 bands) imaging exhibited superior classification results compared to the Canon camera. This superiority was evident across various types of regions of interest and disease stages.

Pérez-Bueno et al. used MLs techniques, such as SVM, artificial neural network (ANN), and logistic regression analysis (LRA), to analyze the avocado crop multispectral images and detect the white root presence [81]. Binary LRA is a commonly employed statistical approach in the field of biomedicine. Its widespread use stems from its ability to estimate the probability of a dichotomous outcome, specifically distinguishing between "healthy" and "infected" conditions. In this paper, the predictive capabilities of NDVI and normalized temperature in the crop canopy were examined as indicators of the disease using the algorithms mentioned above. Among these algorithms, the LRA trained with vegetation index data exhibited better sensitivity and a lower false-positive rate. The models had a high specificity (86.4%), indicating that these models were capable of identifying healthy trees.

Some of the limitations identified in the techniques employed within the studies mentioned in this section include the utilization of a limited amount of training data for certain algorithms and the reliance on controlled conditions for model refinement. These limitations can introduce biases in the practical application of the models in real-world field settings.

### 3.8. Analytical Techniques Applied to Laboratory-Collected Avocado Data

Several of the articles reviewed obtained avocado data from sources other than the crop itself, mainly images of the fruit. Although the ideal is to use as much raw data from the avocado crop as possible, the application of the analytical techniques used in these studies is of interest, especially for replication in the construction of detection tools that can be used outside the laboratory.

Campos Ferreira et al. [83] implemented an ML model using digital images of avocado fruit, through the creation of a CNN classifier, to identify healthy fruit and fruit infected with scab or anthracnose. The fruits were collected at harvest time and taken to the laboratory for imaging. Once trained, the CNN classifier obtained an accuracy of 87% in the detection of fruit diseases, which shows that the use of the model is feasible for the identification of these diseases using digital images at the postharvest stage.

An automated algorithm based on ML to determine the type and quality classification of various fruits and vegetables, including avocados, was proposed by Bhargava et al. [84]. The images are preprocessed using Gaussian filtering to remove noise and improve their quality and then segmented using fuzzy mean clustering and cropping. Then, 114 features are extracted from the images and the feature vector is selected using principal component analysis (PCA). The algorithm with the best results in classifying healthy and defective fruits was SVM, obtaining an accuracy of 96.59%, while the other algorithms evaluated got an accuracy below 90%, showing a promising use of SVM in postharvest applications.

Valiente et al. developed in their study an image processing analysis using non-destructive means to detect defects and maturity of avocados [85]. Using images provided by Google Images for the training of the CNN model, and classifying the type of defect and fruit maturity with this model, they obtained an accuracy of 93% in the identification of defects (anthracnose, insect marks, fruit rot, scarring) and an accuracy of 98.97% in the detection of maturity.

As can be seen, it is evident that ML and DL models are an important option in the optimization of avocado crop production. Yield forecast models can assist growers in crop planning, allowing them to use resources efficiently by analyzing weather conditions, soil characteristics, and historical environmental data. Models that help in pest and disease detection, often using CNNs, allow early identification of abnormalities in avocado trees through image analysis. CNN classifiers offer several advantages in avocado crop analysis. By recognizing patterns and features in avocado images, these models can perform image recognition tasks accurately. The hierarchical feature extraction capability of CNNs allows them to learn complex and abstract representations, making them effective in discriminating subtle visual differences associated with diseases or quality. CNNs are robust to variations in avocado crops thanks to their pooling and convolutional layers, which make them tolerant of spatial variations and small shifts. Additionally, transfer learning can be applied to CNN classifiers, leveraging pretrained models and enhancing their performance even with limited avocado-crop-specific training data. Accordingly, CNN classifiers enhance disease detection, quality assessment, and anomaly identification in avocado crops by providing accurate and automated image recognition, robustness to variability, and the benefit of transfer learning.

*3.9. Summary of Reviewed Studies*

Based on the above subsections, pests in avocados are the most studied problems in the reviewed papers for RQ1, followed by the affectations caused by weather variations and the study of diseases caused by pathogens. However, several studies address two problems simultaneously (pest–climate and disease–climate). Most of them use analytical techniques to estimate the incidence of these biotic and abiotic factors on avocado crops.

Decreased yields caused by abiotic factors (temperature, humidity, precipitation) were the focus of the selected studies for RQ2. Most of the papers studied the effects of water stress on the avocado crop, together with the management strategies that growers apply in the face of a lack of water or poor water quality for their produce. Another biotic factor that has a direct impact on avocado yield is temperature. Both of these factors require adaptive management strategies in the face of climate change in productive zones.

Reviewed studies for RQ3 apply equally in situ and remote sensing techniques for data collection. However, most of the reviewed papers were oriented to the analysis of images of avocado leaves and fruits, and only one study was directly oriented to the use of sensors to capture weather variables data in the avocado crop. All the studies reviewed in this component used analytical techniques that are based on ML and DL algorithms to process avocado data. Among these investigations, the use of CNN and SVM algorithms to analyze crop data corresponds to 22% for each technique (44% between both algorithms), while the other algorithms employed (such as ANN, K-Means, KNN, LRA, MLP, PCA, VAR, Otsu) are used at 7% for each technique (56% between these algorithms). Also, the factors studied and the use cases of sensing and analysis techniques in the papers associated with all research questions were studied and the findings have been compiled and summarized in Table A1.

## 4. Discussion

This paper undertook a systematic review of the scientific literature to establish the current applications of technology in monitoring pests, diseases, and abiotic factors in avocado cultivation and production. The focus of this research was specifically on monitoring APEs in avocado crops, supported by technologies from smart farming.

*4.1. Major Findings and Challenges Encountered*

In the reviewed papers, the use of technology to support management and intervention in avocado crops is highlighted. Thus, several analytical techniques based on ML were used to detect pests in avocado crops, with image-based analysis being the application and/or data format processed with the most research in the studies reviewed.

Papers reviewed and analyzed for RQ1 addressed both abiotic (rainfall, temperature, humidity) and biotic (pests, diseases) factors concerning APEs in avocado crops. The majority of the studies primarily concentrated on the detection and analysis of avocado pests, whereas the research related to disease detection and the identification of climate impacts exhibited more uniformity. The findings demonstrate the influence of endemic pests on both the fruit and leaves of avocado trees, resulting in a decline in the market value of the fruit in one scenario and a reduction in the yield of fruit produced in the other scenario. Progress made in the advancement of prototype sensor systems for insect pest identification, bolstered by ML techniques [86] holds the potential to effectively utilize technology for pest control in avocado farms. By leveraging these technological advancements, avocado farmers can optimize their pest management strategies.

Conversely, papers reviewed and analyzed for RQ2 oriented the impact on yield and risk management in avocado crops. The majority of these studies primarily investigate the adverse consequences stemming from biotic factors (such as water and temperature) that contribute to agroclimatic events. Water scarcity in subtropical regions emerges as a significant factor leading to decreased production of avocado crops, compounded by the inadequate quality of water supplied to the trees. Although water supply is considered secure for avocado crops in tropical regions, climatic variations are expected to affect production in the medium and long term, especially in the main exporting countries [44,67]. This makes future water availability one of the challenges to be addressed, especially to achieve efficient use of this resource in avocado production. Thus, as there is currently widespread water stress in avocado crops in Mexico and Chile, there is a commercial and productive opportunity for countries such as Colombia, given its sufficient water availability and its use without affecting other socio-economic processes. Moreover, the exposure of trees to frost stress or extreme heat becomes a parametric risk that growers must manage by taking timely measures to avoid a decrease in avocado crop yield. Advances in climate predictive methods [87,88] are key to providing tools for growers to take action based on predicted data.

Additionally, papers reviewed and analyzed for RQ3 address analytical techniques and data collection techniques used for pest and disease detection in avocado crops. Most of the studies reviewed used optical sensors (cameras) in various formats to capture data that were then processed using ML algorithms. Another feature is that images that were taken in situ accounted for the majority of the studies, followed by studies that captured data using remote sensing. Advances in the adaptation of ML and DL model-based techniques [89], along with the development of IoT prototypes for crop data capture [21], are the two fronts that should continue to be studied to obtain improvements in avocado production. Among the applications of emerging technologies in agricultural applications, there is a high use of models based on CNN classifiers for the detection of diseases in the leaves of trees, with research working on the preparation of these models through transfer learning to detect diseases in products such as corn, apples, and tomatoes, with accuracy measurements between 98.6% and 99.4% obtained in the testing of classifiers [90–92]. These advances are relevant for the use of emerging technologies in the identification of APEs, and in the specific case of avocado crops, the use of data from the crop and the use of pretrained models with related data should be prioritized since there is a bias in the application of transfer learning to pretrained models for other agricultural products. Characteristics such as tree phenotyping, leaf insertion angle on plants, and canopy distribution are important for both data collection and training of AI-based models.

An important advancement that has emerged is the incorporation of multispectral imaging methods, which offer comprehensive spectral data for enhanced detection and categorization of pests and diseases. Using these techniques, it can detect subtle changes in tree health and identify early signs of pest infestation and disease. Moreover, the research highlights the importance of integrating and combining data from various sources, such as sensors, weather stations, and satellite imagery, to enhance the precision and accuracy of pest and disease detection models. Avocado crop data from diverse sources can help

avocado producers gain a fuller understanding of how environmental factors, crop health, and pest dynamics interact. To facilitate effective decision-making and crop monitoring, the integration of ML and DL models with IoT prototypes has become a challenging task. IoT-based sensor networks enable continuous data collection, enabling farmers to receive timely alerts and take proactive measures against pests and diseases. These advancements not only improve the efficiency of pest management practices but also help optimize resource allocation and minimize the use of chemical pesticides.

The reviewed papers revealed certain limitations regarding data collection from avocado crops and the specificity of avocado-related data. A majority of the data collection was conducted under controlled laboratory conditions, which restricts the application of study findings to real-world crop conditions, particularly in agricultural production areas. Consequently, one of the critical challenges for future research is to parameterize the utilization of technologies in field conditions and to gather data directly from agricultural fields. Addressing this challenge would involve conducting studies in actual avocado production areas, where data can be collected under real-world agricultural conditions. This would provide more accurate insights into the performance and effectiveness of various technologies in practical farming scenarios.

*4.2. Future Outlook and Research Trends*

Future research and work should focus on the development of ML and DL models applied to monitoring and management of the risk associated with pest and weather variations in avocado production. Using the crop data, it becomes feasible to customize pretrained models and establish parameters for predicting the impact of pests and weather conditions on yield performance. Consequently, future research endeavors should prioritize the successful implementation of smart farming practices within avocado cultivation. This entails conducting a comprehensive cost-versus-efficiency analysis of the technologies employed for monitoring and detecting APEs. Moreover, it is highly recommended to promote the adoption of 4IR technologies to enhance avocado production, particularly in countries with emerging economies where avocado holds strategic significance as an agricultural commodity for economic growth.

*4.3. Limitations of This SLR*

This study is subject to certain limitations. The literature search was conducted using three scientific databases: Web of Science, Scopus, and IEEE Xplore. While these databases encompass various domains and include multiple individual databases, it is important to acknowledge that the exclusion of other databases may have resulted in the omission of potentially relevant papers from the review. The search strategy employed in this study could have potentially impacted the number of papers included in the analysis. Factors such as the specific search string used and the decision to focus on papers published within the last six years may have influenced the selection of relevant papers. While it is acknowledged that these limitations may have affected the total number of papers obtained and considered in the review, it is affirmed that these limitations did not significantly impact the overall discussion and conclusions drawn from the included papers. The study's findings and interpretations should be viewed within the context of these limitations, and future research could consider incorporating additional databases and broader search criteria to ensure a more comprehensive coverage of the relevant literature.

## 5. Conclusions

A total of 37 papers which satisfied the eligibility criteria for the research questions posed were reviewed in this study. This review revealed that the primary focus of technological applications in avocado crops lies in the detection of pest insects, which represent the most significant biotic factor influencing phytosanitary events. Furthermore, efforts have been directed towards addressing the effects of water deficit, extreme temperatures, and excess humidity on avocado crops, as these factors are the major contributors to agro-

climatic events. Several ML algorithms were utilized in the analyzed studies for data analysis purposes, with SVMs and CNNs algorithms as the most frequently employed techniques. Moreover, in terms of data collection technique usage, images collected with in situ devices (such as portable cameras prototypes) were found to be the most commonly utilized sensors in the reviewed papers.

**Author Contributions:** Conceptualization, T.R.-G. and M.I.H.-P.; methodology, T.R.-G.; validation, T.R.-G., M.I.H.-P., M.S.T., A.M.-T., E.V. and A.P.; formal analysis, T.R.-G.; investigation, T.R.-G.; resources, T.R.-G., M.I.H.-P., M.S.T. and E.V.; data curation, T.R.-G.; writing—original draft preparation, T.R.-G.; writing—review and editing, T.R.-G., M.I.H.-P., M.S.T., A.M.-T., E.V. and A.P.; visualization, T.R.-G.; supervision, M.I.H.-P.; project administration, M.I.H.-P.; funding acquisition, M.I.H.-P. All authors have read and agreed to the published version of the manuscript.

**Funding:** This research was funded by Universidad EAFIT, Vicerrectoría de Ciencia, Tecnología e Innovación, research project 1111-11110022021.

**Institutional Review Board Statement:** Not applicable.

**Informed Consent Statement:** Not applicable.

**Data Availability Statement:** Not applicable.

**Acknowledgments:** The authors thank the Vicerrectoría de Ciencia, Tecnología e Innovación of Universidad EAFIT for funding this research project. In addition, T.R.-G. thanks the Universidad EAFIT for his Ph.D. scholarship and extends his thanks to Mauricio Toro, Leidy Marcela Dueñas Ramirez, and the undergraduate students of the Semillero AgroTech for their constructive contributions during the methodological stages.

**Conflicts of Interest:** The authors declare no conflict of interest.

## Abbreviations

The following abbreviations are used in this paper:

| | |
|---|---|
| APE | Agroclimatic and Phytosanitary Event |
| 4IR | Fourth Industrial Revolution |
| IT | Information Technology |
| IoT | Internet of Things |
| IIoT | Industrial Internet of Things |
| AI | Artificial Intelligence |
| ML | Machine Learning |
| DL | Deep Learning |
| BD | Big Data |
| CPS | Cyber-Physical System |
| SLR | Systematic Literature Review |
| PRISMA | Preferred Reporting Items for Systematic Reviews and Meta-Analyses |
| RQ | Research Question |
| IC | Inclusion Criteria |
| EC | Exclusion Criteria |
| NVDI | Normalized Difference Vegetation Index |
| FNN | Fuzzy Neural Network |
| MaxEnt | Maximun Entropy Method |
| VAR | Vector Autoregressive Modeling |
| CNN | Convolutional Neural Network |
| MLP | Multi-Layer Perceptron |
| KNN | K-Nearest Neighborhood |
| SVM | Support Vector Machine |
| ANN | Artificial Neural Network |
| LRA | Logistic Regression Analysis |
| UAV | Unmanned Aerial Vehicle |
| PCA | Principal Component Analysis |

## Appendix A

Table A1 contains the factors studied that have impacts on avocado crops, as well as the use cases of APE detection and analysis techniques in the papers reviewed in this SLR, which are associated with all the research questions raised.

**Table A1.** Papers distributed by factors studied, sensing, and analysis techniques.

| Ref. | Factor Studied | | Sensing Technique | | | Analysis Technique | |
|------|---------|--------------|---------|--------|-----|------------|----------------|
| | Climate | Pest/Disease | In Situ | Remote | Lab | Regression | Classification |
| [44] | X | | | | X | X | |
| [45] | X | | X | | | | X |
| [48] | X | X | X | | | | |
| [49] | | X | X | | X | | X |
| [50] | | X | X | | | | X |
| [51] | | X | X | | | | X |
| [52] | | X | X | | | | X |
| [53] | | X | X | | | X | |
| [54] | | X | | | X | X | |
| [56] | | X | X | | | X | |
| [57] | X | X | X | | | X | |
| [58] | X | X | X | | | X | |
| [59] | X | X | X | | X | X | |
| [60] | | X | X | | | X | |
| [61] | | X | | | X | | |
| [62] | | X | | | | | |
| [63] | X | | | X | | | X |
| [64] | X | | X | | | X | |
| [65] | X | | X | | | X | |
| [66] | X | | X | | | X | |
| [67] | X | | | | X | | X |
| [68] | X | | | | X | X | |
| [69] | X | | X | | | X | |
| [70] | | | X | | | X | |
| [71] | | X | X | | | | X |
| [77] | X | | X | | | | X |
| [78] | | X | X | | | | X |
| [80] | | X | | X | | | X |
| [43] | | X | | X | | | X |
| [81] | | X | | X | | X | |
| [82] | X | | | X | | | X |
| [79] | | X | X | | | | X |
| [83] | | X | X | | X | | X |
| [84] | | | X | | | | X |
| [85] | | X | | | X | | X |

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
