# Peer review of "Agroclimatic and Phytosanitary Events and Emerging Technologies for Their Identification in Avocado Crops: A Systematic Literature Review"

_agronomy, doi:10.3390/agronomy13081976_

Round 1

Reviewer 1 Report

This manuscript presents a systematic literature review (SLR) about the use of technologies for monitoring phytosanitary (pests, diseases) and agroclimatic (weather variables) agents in avocado crops. Challenges and trends in the parameterization of the technology in field conditions for data collection were also highlighted.

 Line 176: It seems to be wrong, the word "o", or "or"?

Line 144696 and Figure 1: The word "IEEEXplore" should be changed to "IEEE Xplore".

What are the other sources mentioned in Line 164-165?

The information in Figure 2 seems to have been included in Figure 3, is there still a point to Figure 2?

In Chapter 3, the authors elaborate on the work of other researchers in this area and the achievement. However, in order to enhance the logical expression between each paragraph, the use of simple words for splicing should be avoided.

The second paragraph of the conclusion is the future outlook and should be placed in the discussion section

Reviewer 2 Report

- The number of reviewed papers is very small. 

- The crux of the paper is focused on how the survey/review was conducted, which is part of the PRISMA standard process. However, the authors should foucs on the outcomes of such process rather than describing it. In other words, there should be more depth in identifying the research landscape, problems, and technological solution. Also, there needs to be an in depth discussion of future research directions. 

- I am not sure if it is necessary to list the number of papers per publisher especially that most are 1. The same for Figure 3. 

- Figure 6, what is the utility of such figure. 

- Similar studies for diseases identification and classification in a very similar fashion to Avocado crops can be discussed, see Fraiwan M, Faouri E, Khasawneh N. Classification of Corn Diseases from Leaf Images Using Deep Transfer Learning. Plants (Basel). 2022 Oct 11;11(20):2668. doi: 10.3390/plants11202668. PMID: 36297692; PMCID: PMC9609100.

Reviewer 3 Report

The authors conducted a system review of agroclimatic and phytosanitary events and techniques in avocado. The topic is interesting. However, I have some suggestions as follows.

1. Figure 2 is redundant and can be combined with Figure 3 for better clarity and presentation.

2. The paper references multiple works. It is crucial for the authors to assess the confidence of the results presented in these papers, particularly for those focusing on correlations. To enhance the analysis, it is suggested that the authors evaluate potential sources of bias, such as biases arising from the randomization process, missing outcome data, measurement of outcomes, and selection of reported results.

3. In Section 3.3.2, further analysis of the analytic techniques is necessary. The authors should provide a more detailed explanation of the evolution and future development of these techniques. Specifically, they should discuss the primary machine learning (ML) models used in the field and highlight the challenges faced in algorithms and computational methods to mitigate problems in the avocado industry. For example, in image processing, it would be beneficial to describe the methods used prior to convolutional neural networks (CNN) and elaborate on the advantages of CNN over other image processing models.

4. The abstract mentions the adoption of Industry 4.0 technologies. In Section 3.3.1, it is important to emphasize the concept of Industry 4.0 in agriculture and provide a clear definition. Additionally, the authors should select relevant standards to measure the platforms instead of listing them individually.

5. In Section 3.3.1, it is recommended to decouple the review of data collection techniques from the review of analytical techniques.

6. The review of ML methods and IoT techniques appears to be shallow overall. To enhance the quality of the review, the authors should provide more in-depth analysis and discussion of these methods.

Round 2

Reviewer 2 Report

The authors addressed my comments.